# Mitochondrial E3 Ubiquitin Ligase Parkin: Relationships with Other Causal Proteins in Familial Parkinson’s Disease and Its Substrate-Involved Mouse Experimental Models

**DOI:** 10.3390/ijms21041202

**Published:** 2020-02-11

**Authors:** Satoru Torii, Shuya Kasai, Tatsushi Yoshida, Ken-ichi Yasumoto, Shigeomi Shimizu

**Affiliations:** 1Department of Pathological Cell Biology, Medical Research Institute, Tokyo Medical and Dental University (TMDU), 1-5-45 Yushima, Bunkyo-ku, Tokyo 113-8510, Japan; 2Department of Stress Response Science, Center for Advanced Medical Research, Graduate School of Medicine, Hirosaki University, 5 Zaifu-cho, Hirosaki, Aomori 036-8562, Japan; 3Department of Biochemistry and Molecular Biology, Graduate School of Medical Science, Kyoto Prefectural University of Medicine, Kawaramachi-Hirokoji, Kamigyo-ku, Kyoto 602-8566, Japan; 4Department of Molecular and Chemical Life Sciences, Graduate School of Life Sciences, Tohoku University, Sendai 980-8578, Japan

**Keywords:** Parkinson’s disease, 1-methyl-4-phenyl-1,2,3,6-tetrahydropyridine (MPTP), Parkin, PINK1, HtrA2/Omi, IPAS

## Abstract

Parkinson’s disease (PD) is a common neurodegenerative disorder. Recent identification of genes linked to familial forms of PD has revealed that post-translational modifications, such as phosphorylation and ubiquitination of proteins, are key factors in disease pathogenesis. In PD, E3 ubiquitin ligase Parkin and the serine/threonine-protein kinase PTEN-induced kinase 1 (PINK1) mediate the mitophagy pathway for mitochondrial quality control via phosphorylation and ubiquitination of their substrates. In this review, we first focus on well-characterized PINK1 phosphorylation motifs. Second, we describe our findings concerning relationships between Parkin and HtrA2/Omi, a protein involved in familial PD. Third, we describe our findings regarding inhibitory PAS (Per/Arnt/Sim) domain protein (IPAS), a member of PINK1 and Parkin substrates, involved in neurodegeneration during PD. IPAS is a dual-function protein involved in transcriptional repression of hypoxic responses and the pro-apoptotic activities.

## 1. Introduction

Parkinson’s disease (PD) is the second most common neurodegenerative disease and is characterized by progressive resting tremors, rigidity, bradykinesia, gait disturbances, postural instability, and dementia [1,2]. The motor symptoms of PD are associated with the degeneration of dopaminergic neurons in the substantia nigra pars compacta (SNpc). Although most PD cases are sporadic, approximately 10% are familial cases [3]. The pathogenesis of sporadic PD is yet to be established, but it is suggested that genetic predispositions and environmental toxins causing mitochondrial dysfunction and oxidative stress are involved [1,2,3]. In the early 1980s, the first evidence linking PD to mitochondrial dysfunction appeared with the observation that exposure to 1-methyl-4-phenyl-1,2,3,6-tetrahydropyridine (MPTP) causes dopaminergic neurodegeneration [4]. MPTP is a synthetic by-product of meperidine production. The product of MPTP oxidation, 1-methyl-4-phenylpyridinium (MPP^+^), selectively enters dopaminergic neurons via the dopamine transporters and inhibits mitochondrial complex I, a key component of the mitochondrial respiratory chain [5]. Subsequently, MPP^+^ induces oxidative stress and dopaminergic neuronal death [6]. Exposure to other toxins, including 6-OHDA, or the pesticides rotenone and paraquat, also results in the loss of nigrostriatal dopaminergic neurons [7,8]. They produce effects similar to parkinsonian phenotypes in animal models and humans. In MPTP-induced mouse models, there are two typical dosing regimens for MPTP, acute and subacute [9]. In the former type, mice are intraperitoneally injected four times with MPTP (15–20 mg/kg) at 2-h intervals (6-h period) within a single day. Dopaminergic neuron loss can be observed at 3–7 days after the last dose of MPTP. For the latter type, mice were intraperitoneally injected once a day with MPTP (30 mg/kg) for five consecutive days. Dopaminergic neuron loss can be observed 7 days after the last dose of MPTP. The concentrations of MPTP and MPP^+^ in plasma, striatum, and cortex were studied in mice [10]. Elevated concentrations of MPP^+^ in plasma and the striatum are maintained for 72 h and 12 h after administration, respectively. N-methylmercaptoimidazole increases the amount of MPTP delivered from the peripheral nervous system to the central nervous system [10].

Accumulated genetic analyses have identified that several gene mutations are associated with an early onset of parkinsonism, including *α-synuclein* (*SNCA*), (*PARK1* and *PARK4*), *Parkin* (*PARK2*), *PTEN-induced kinase 1* (*PINK1*) (*PARK6*), *DJ-1* (*PARK7*), *leucine rich repeat kinase 2* (*LRRK2*) (*PARK8*), *ATP13A2* (*PARK9*), *HtrA2/Omi* (*PARK13*), *PLA2G6* (*PARK14*), *VPS35* (*PARK17*), *coiled-coil-helix-coiled-coil-helix (CHCH) domain 2* (*CHCHD2*) (*PARK22*) [3,11]. Among these, mutations of park genes including *Parkin*, *PINK1*, *HtrA2/Omi* and *CHCHD2* are directly involved in mitochondrial dysfunction [12,13,14,15]. PINK1, HtrA2/Omi, and CHCHD2 proteins have a positively charged mitochondrial targeting sequence (MTS) in their N-terminus that forms an amphipathic α-helix (Figure 1). Meanwhile, Parkin does not have an MTS, but does translocate to the mitochondria during mitochondrial dysfunction (Figure 1). Studies of these genes using knockout mice and knockout cell lines demonstrate a relationship between these proteins and mitochondrial quality control [12,13,14,15,16]. Autosomal recessive mutations in the *Parkin* gene (*PARK2*), which encodes an E3 ubiquitin ligase, are the most common cause of early onset of PD [1,2,3]. Recently, increased in vitro studies examining the molecular mechanisms of the PINK1/Parkin pathway using the mitochondrial uncouplers, carbonyl cyanide *m*-chlorophenyl hydrazone (CCCP) and carbonyl cyanide *p*-(trifluoromethoxy) phenyl hydrazone (FCCP) [13,14]. In healthy mitochondria, PINK1 is translocated and imported to mitochondria via its N-terminal MTS. There, it undergoes cleavage by the mitochondrial processing peptidase (MPP) in the inner membrane to form a 60 kDa intermediate [17]. Following, a presenilin-associated rhomboid-like protein (PARL) and/or matrix-AAA (*m*-AAA) proteases cleave it to generate a 52 kDa processed form of PINK1 that attached to the inner membrane [17,18]. The cleavage site by PARL has been mapped to Ala^103^, near the MTS in PINK1 [17,19]. N-terminally cleaved PINK1 is then degraded by the ubiquitin/proteasome system in cytosol. Under these conditions, Parkin remains inactive in the cytosol [13]. Upon mitochondrial damage, such as oxidative stress and experimental CCCP treatment, the membrane potential of mitochondria is decreased; hence, PINK1 is not imported into mitochondria and accumulates on the outer mitochondrial membrane (OMM) to activate Parkin [13,18]. PINK1-mediated phosphorylation of the ubiquitin and the ubiquitin-like (UBL) domain of Parkin enable its E3 ubiquitin ligase functions in concert with E2 ubiquitin-conjugating enzymes [20,21,22,23]. Poly-ubiquitin with p-Ser^65^ bound to OMM substrates acts as a Parkin receptor; further recruiting Parkin to the mitochondria. These poly-ubiquitin chains can be cleaved by de-ubiquitinating enzymes, including USP15 and USP30, to reverse PINK1/Parkin functions [24,25]. Individual OMM proteins decorated with poly-ubiquitin can be extracted from the mitochondrial membrane and degraded by the 26 S proteasomes. Severely damaged mitochondria are decorated by large numbers of poly-ubiquitin chains that signal for the accumulation of the ubiquitin-binding cargo receptors, NDP52, optineurin, and p62 [26]. NDP52 can bind LC3 in phagophores and recruit the autophagy initiator complex, Fip200-Ulk1, leading to progression of the mitochondrial autophagy (mitophagy) [26,27]. Poly-ubiquitin chains also serves as tags for the accumulation of RabGEF1 on mitochondria, leading to activation of Rab5-Rab7 signaling and the subsequent recruitment of Atg9A vesicles to mitochondria [28]. Autophagosomes containing damaged mitochondria fuse with lysosomes, leading to the generation of autolysosomes, where the cargo is degraded [26]. Recently, our group developed a mitochondria-selective pH-sensitive small-molecule fluorescent dye, Mtphagy Dye, which is incorporated into mitochondria and emits a strong red fluorescence in autolysosomes upon acidification during mitophagy [29]. Mtphagy Dye is a useful tool for easy visualization and detection of mitophagy.

PINK1 was initially named PTEN-induced “putative” kinase-1, and the topology of PINK1 on/in mitochondria was unknown. Although tumor necrosis factor type 1 receptor–associated protein (TRAP1) and the serine protease HtrA2/Omi have been identified as putative PINK1 targets [30,31], TRAP1 and HtrA2/Omi are localized primarily in the mitochondrial matrix and mitochondrial intermembrane space, respectively. Furthermore, previous reports suggest that the kinase domain of mitochondrial PINK1 faces the cytoplasm [32]. Subsequently, increasing numbers of studies using CCCP treatment have investigated the substrates of PINK1 phosphorylation, including Miro1/Rhot1 [33], PINK1 (autophosphorylation) [34], Parkin [20,21], Mitofusin 2 [35], Ubiquitin [22,23], inhibitory PAS (Per/Arnt/Sim) domain protein (IPAS) [36], Parkin-Interacting Substrate (PARIS) [37], and drosophila MIC60/Mitofilin [38]. In addition to LC-MS/MS phosphorylation analyses, some studies used Phos-tag gel mobility shift assays, a powerful and useful tool for detecting the phosphorylation of proteins [39]. Our group has reported well-characterized PINK1 phosphorylation motifs [36] and updates them in this review (Figure 2). Almost all PINK1 target sites in *Mus musculus* have hydrophobic amino acids at position +2 after the phosphorylation site. In addition, hydrophobic amino acids or acidic amino acids at position +1 after phosphorylation sites are preferred by PINK1. Basic amino acids at positions −3 and −2 (in some cases −5, −4 and −1) are preferred, expect for Parkin. The phosphorylation sites of PARIS are Ser^322^-P and Ser^613^-P. Both serine sites have a proline at position +1. As a pS/T-P sequence is preferred by mitogen activated protein (MAP) kinases and cyclin-dependent kinases (CDK), the phosphorylations of PARIS could be regulated by MAP kinases similar to HtrA2/Omi (chapter 2). Thus, PINK1 phosphorylation consensus motifs are valuable for predicting the phosphorylation sites of PINK1-binding proteins and bioinformatical approaches for detecting mitochondrial PINK1-target proteins.

As mentioned above, many studies report the activation mechanism of the PINK1/Parkin pathway and the pathogenesis of PD using cultured cells and mouse models. Next, we will review our findings including the relationships between Parkin and HtrA2/Omi in Parkin-transgenic (Parkin-Tg) mice [40].

## 2. Parkin-Tg Introduction into mnd2 Mice; Neurodegeneration in HtrA2/Omi Mutant Mice Is Not Rescued by Parkin Transgene Expression

HtrA2/Omi is a serine protease localized in mitochondria (Figure 1) [12]. HtrA2/Omi is expressed as a 49 kDa proenzyme that is targeted to the intermembrane space of the mitochondria where it undergoes proteolytic maturation via cleavage of the first 133 N-terminal residues [41]. HtrA2/Omi contains an N-terminal trypsin-like protease domain and a C-terminal PDZ domain (Figure 1). Ser^306^ is the active site of the protease domain and mutation of Ser^306^ inactivates HtrA2/Omi [12]. Ser^142^ and Ser^400^ are the known phosphorylation sites of HtrA2/Omi [30,42]. Upon oxidative stress, p38 MAP kinase phosphorylates Ser^142^ of HtrA2/Omi in a PINK1-depdenent manner. CDK5 phosphorylates Ser^400^ of HtrA2/Omi in a p38-dependent manner, resulting in the enhancement of HtrA2/Omi proteolytic activity and increased resistance of cells to mitochondrial stress. Meanwhile, mature HtrA2/Omi is released from mitochondria into the cytosol where it is bound to the inhibitor of apoptosis proteins (IAPs); promoting cell death upon apoptotic induction [43,44]. HtrA2/Omi has therefore been proposed to be a pro-apoptotic protein. However, more importantly, HtrA2/Omi is required for maintaining mitochondrial function. This opinion is supported by studies employing HtrA2/Omi-deficient mice, motor neuron degeneration mutant (mnd2) mice [12]. Mnd2 mice possess a non-functional missense mutation at Ser^276^ to Cys in the protease domain of HtrA2/Omi, close to Ser^306^ [12]. Protease activity of HtrA2/Omi is greatly reduced in the tissues of mnd2 mice. Mnd2 mutant cells are more vulnerable to Ca^2+^-induced permeability transitions and mitochondrial membrane permeabilization [12]. Mnd2 mice exhibit weight loss, muscle wasting, neurodegeneration, involution of the spleen and thymus, and finally die within 40 days of birth, and therefore are considered to be a useful animal model for PD [45]. Neuronal cell death in mnd2 mice exhibits features of both apoptosis and necrosis, whereas motor abnormalities and striatal neuronal loss in mnd2 mice are not prevented by overexpression of the anti-apoptotic protein Bcl-2 [46] or cyclophilin D deficiency [47]. Interestingly, Parkin overexpression prevents both neurotoxin-induced and mutant α-synuclein-induced PD models [48,49,50]. Injection of lentiviral vector encoding human Parkin has neuroprotective effects in 6-hydroxydopamine (6-OHDA) rat models for PD; rats overexpressing Parkin display behavioral improvements [48]. Parkin expression protects dopaminergic neurons against the toxicity associated with mutant α-synuclein in vitro [49] and in vivo [50]; thus, Parkin deficiency and α-synuclein mutations are linked to the common pathogenic symptoms associated neuronal cell death. In addition, Parkin overexpression protects neurons from tau-induced dopaminergic degeneration in rats [51]. As mentioned above, both Parkin and PINK1 play a role in the elimination of damaged mitochondria by mitophagy [14], and these functions may be the mechanism by which Parkin protects against PD. With this hypothesis in mind, we examined the expression levels of these proteins in mnd2 mouse brains [40]. Parkin protein levels in the striatum were dramatically reduced in the mnd2 mice at 4 weeks after birth. The decrease started from 2 weeks after birth before neurodegenerative symptoms were observable. Although Parkin protein levels in hippocampus were also decreased in mnd2 mice, the decrease was weaker than that in striatum. The protein levels of α-synuclein, another PD-related protein, were not changed in mnd2 mice when compared to wild-type (WT) littermates. These results suggest that the decrease of Parkin expression may be involved in the neurodegenerative disorders of mnd2 mice (Figure 3) [40], and we hypothesize that when Parkin expression is compensated for by overexpression, neurodegeneration can be delayed or recovered. Thus, we generated Parkin-Tg mice where Parkin was specifically expressed in the brain by regulation of a prion promoter [40]. We observed increased levels of Parkin in striatum, hippocampus, and cerebral cortex in Parkin-Tg mice. Parkin-Tg mice have no remarkable abnormalities or neurodegenerative symptoms. Parkin-Tg mice were then crossed with mnd2 mice to generate Parkin-Tg/mnd2 mice. Both Parkin-Tg/mnd2 mice and mnd2 mice were smaller than WT mice. The body weight of Parkin-Tg/mnd2 and mnd2 barely increased over the first 2 weeks after birth and was about 50% of WT mice after 4 weeks. Body weight trends in mnd2 and Parkin-Tg/mnd2 mice were almost identical. Previous reports indicate that mnd2 mice died within 40 days after birth. Therefore, we examined whether the Parkin transgene prolongs the survival rate of mnd2 mice. However, the average survival of mnd2 and Parkin-Tg/mnd2 mice was the same (mnd2: 30.5 ± 9.14 days and Parkin-Tg/mnd2: 27.8 ± 8.16 days). To examine behaviors reflecting neuromuscular abnormalities, hanging wire tests were performed. Mice need balance and grip strength to remain on the wire net during the test. WT mice remained on the wire net for longer than 60 s, while mnd2 mice and Parkin-Tg/mnd2 mice dropped within 17 s. These results indicate that the Parkin transgene does not rescue the disabilities of mnd2 mice [40]. Taken together, although Parkin protein levels in the striatum is decreased in mnd2 mice, the Parkin transgene did not eliminate the neurodegenerative disorders in the mnd2 mice (Figure 3). However, the observation that Parkin protein levels are significantly decreased in striatum of mnd2 mice is an interesting finding; this suggests that HtrA2/Omi controls the transcriptional level of Parkin mRNA and/or the stability of Parkin proteins due to post-translational regulation. The latter is more likely, because HtrA2/Omi is a protease that can cleave other proteins involved in Parkin removal. Further studies are necessary to elucidate the mechanism(s). As the Parkin transgene does not eliminate the neurodegenerative disorders in the mnd2 mice, either the Parkin protein does not play a major role during neurodegeneration in mnd2 mice or the Parkin transgene alone is not sufficient to rescue mitochondrial dysfunction. In latter pattern, PINK1 up-regulation is necessary for further activation of Parkin.

## 3. One of PINK1/Parkin-Target Proteins, IPAS; Its Molecular Mechanism of Pro-Apoptotic Activation, and Its Deficient Mouse Model

Some reports have identified multiple Parkin substrates [52], such as the Pael receptor [53], Miro1/Rhot1 [33], FAF1 [54], Mitofusin 2 [35], IPAS [36], BNIP3L [55], PARIS [37], and MITOL/March5 [56]. These Parkin substrates are also phosphorylated by PINK1 which induces Parkin-mediated ubiquitination. Parkin interacts with and ubiquitinates these proteins, leading to their degradation, except for MITOL/March5. Parkin-mediated ubiquitination of MITOL/March5 causes its translocation to peroxisomes [56]. As mentioned above, MPTP inhibits mitochondrial complex I activity, resulting in an increased reactive oxygen species generation. In addition, MPTP inhibits the ubiquitin ligase activity of Parkin due to *S*-nitrosylation in vitro and in vivo [57]. Hence, MPTP is useful for experimental PD mouse models (for relating Parkin substrates) and Parkin substrates accumulate in the SNpc of MPTP-treated mice and their knockout mice are protected from MPTP-induced neuronal degeneration [36,54,58]. This review focuses on our findings regarding IPAS, a substrate of both PINK1 and Parkin enzyme activity [36].

IPAS was first reported as a potent negative regulator of hypoxia-inducible factor-1 (HIF-1) [59], a master regulator of mammalian oxygen homeostasis [60]. IPAS directly binds to HIF-1α, an oxygen-sensitive subunit of HIF-1, preventing their binding to hypoxia response elements (HRE) localized in the transcription-control region of subordinate genes. The binding of IPAS to HIF-1α-like factor (HLF, also known as HIF-2α and EPAS1) has also been observed. IPAS is one of the alternatively spliced variants of HIF-3α, a family member of HIF-1α [61]. Expression of IPAS is highly tissue specific. Makino et al. found that IPAS is constitutively expressed in the Purkinje cells of the cerebellum and corneal epithelium where IPAS was involved in the regulation of angiogenesis [59]. Our observation strongly suggested that IPAS is inducibly expressed in the neurons but not in glial cells of the central nervous system [36]. To our knowledge, IPAS is expressed only in PC12 cells but no other cell culture lines, suggesting the expression of IPAS to be tissue specific [62]. In addition, IPAS is induced by both sustained hypoxia and intermittent hypoxia-induced oxidative stress/NF-κB signaling via the activation of HIF-1 [59,62]. Once induced, IPAS localizes to the nucleus and functions as a negative feedback inhibitor of HIF family-dependent hypoxic responses (Figure 4A). As induction of IPAS is also dependent on the TNF-alpha/NF-κB signaling pathway, there is crosstalk between inflammation pathways and HIF-dependent hypoxic responses [63]. We reported a novel function of IPAS as a pro-apoptotic factor acting on mitochondria [64]. Detailed analyses using subcellular localization studies demonstrate that IPAS is localized to both the nucleus and mitochondria. IPAS has a bipartite-like nuclear localization signal and nuclear export signal in its N- and C-terminal region, respectively, and may shuttle between the nucleus and cytosol [65]. Other HIF-3α variants have nuclear localization signal, but they neither have nuclear export signal nor shuttle between the nucleus and cytosol [66,67]. Mitochondrial IPAS interacts with pro-survival Bcl-2 proteins, including Bcl-x_L_, Bcl-w, and Mcl-1, but not Bcl-2 and A1a, through its C-terminal domain (CTD). The conformation change in IPAS occurs by binding of Bcl-x_L_ [68]. The physical interactions between IPAS and pro-survival factors results in the dissociation of Bax from pro-survival factor and facilitates the translocation of Bax to mitochondria. Bax translocation leads to the release of cytochrome c from the mitochondria, activation of caspase-3, and finally apoptosis. These findings demonstrate that IPAS is a dual-function protein involved in both transcription repression and apoptosis (Figure 4A), although other HIF-3α variants are involved in transcription regulation [66].

We have sought post-translational modifications of IPAS, by which the pro-apoptotic activities of IPAS may be regulated. We observe that the phosphorylation of IPAS Ser^184^ by MAP kinase-activated protein kinase 2 (MK2 or MAP-KAPK2), a direct substrate of p38 MAP kinase, enhances the pro-apoptotic activity of IPAS (Figure 4A) [69]. Perinuclear clustering of mitochondria and activation of caspase-3 induced by IPAS expression increased by ultraviolet B irradiation (312 nm), which activates the p38 MAP kinase cascade; this enhancement was completely blocked by a p38 MAP kinase inhibitor, a MK2 inhibitor, and siRNA treatment against MK2. Mass spectrometry analysis revealed that several serine residues including Ser^184^, Ser^219^, and Ser^223^ in the CTD of IPAS are phosphorylated. Of these serine residues, Ser^219^ and Ser^223^ match with the p38 MAP kinase consensus sequence, PXpS/TP. The minimal consensus for MK2 phosphorylation has been reported to be RXXpS/T [70]; Ser^184^ (RMKpS) completely matches with this consensus motif. Among these identified phosphorylation sites, replacement of Ser^184^ by Ala alone led to a total loss of enhanced perinuclear clustering of mitochondria and the activation of caspase-3 in UVB-irradiated cells. Furthermore, replacement of Ser^184^ by Asp, a phosphomimetic mutation, led to the enhancement of abnormality of mitochondria and the activation of caspase-3 without UVB irradiation. Notably, the interaction between the IPAS phosphomimetic mutant and Bcl-x_L_ was up-regulated, indicating the phosphorylation of IPAS Ser^184^ enhances its affinity for pro-survival Bcl-2 family proteins. Collectively, the pro-apoptotic activity of IPAS is regulated by the p38 MAP kinase/MK2 signaling pathway (Figure 4A).

Finally, we investigated physiological and pathological processes in which the pro-apoptotic activity of IPAS is involved. Considering that IPAS is induced by oxidative stress in neural tissues and that the pro-apoptotic activity of mitochondrial IPAS is activated by oxidative stress, we suggest a relationship between IPAS-induced neural cell death and neurodegenerative disorders. PD researchers have identified mitochondrial targets for both PINK1 phosphorylation and Parkin ubiquitination, such as Miro1/Rhot1 and Mitofusin2. Therefore, we hypothesize that IPAS is also a good mitochondrial substrate for PINK1 and Parkin. Our studies confirmed this hypothesis [36,71]. When cultured cells (SH-SY5Y and HeLa cells) were treated with CCCP, in the presence of MG132 to inhibit proteasome activity, mitochondrial IPAS was bound to Parkin via its CTD. IPAS is subsequently ubiquitinated by Parkin at the position lys^167–169^ in the second PAS-like domain (Figure 4B). The poly-ubiquitin chains of IPAS are predominantly K48 linkages, which are known as a tag for proteasomal degradation of target proteins. K63 linkages on IPAS were not observed after CCCP treatment. The degradation of mitochondrial IPAS, but not nuclear IPAS, was rapidly increased by CCCP treatment. Pathogenic missense mutations of Parkin present in RING0, RING1, and IBR, and RING2 domains exhibited a little or no ubiquitination activity for IPAS. However, the R42P mutant of Parkin enhances IPAS ubiquitination for unknown reasons. Perhaps, Parkin (R42P) may enhance interactions with substrates, as IPAS is markedly bound to Parkin (R42P) without CCCP treatment. Of note, CCCP treatment also increases the interactions between IPAS and PINK1. Moreover, IPAS is phosphorylated by PINK1 at several sites including Thr^12^. The sequence flanking Thr^12^ closely matches with the PINK1 phosphorylation consensus motifs (Figure 2). PINK1 siRNA treatment reduces both IPAS phosphorylation and the binding of IPAS to Parkin. Moreover, replacement of Thr^12^ by Ala leads to reduced interactions between IPAS and Parkin as well as a total loss of IPAS ubiquitination by Parkin; suggesting that IPAS phosphorylation by PINK1 induces conformation changes in IPAS to make its PAS-like domain accessible to Parkin. Treatment of cells with CCCP completely abolished IPAS-induced apoptosis; notably, the inhibition of apoptosis was reversed by a concomitant knockdown of Parkin. Parkin WT, but not a ligase-deficient mutant (T415N), also decreases the IPAS-induced apoptosis. Collectively, these results strongly suggest that IPAS is a mitochondrial substrate in the PINK1-Parkin pathway and is stabilized when PINK1-Parkin activity is decreased. Up-regulation of IPAS leads to apoptosis in neural cells (Figure 4B) [36].

We next sought evidence for causal relationship between IPAS expression and PD pathogenesis. IPAS mRNA and protein are rapidly and strongly induced after at four injections of MPTP. Immunohistochemical analysis indicate that induced IPAS proteins are expressed mainly in the cytoplasm region, but not in nucleus, of tyrosine hydroxylase (TH)-expressing dopaminergic neurons in the SNpc. Therefore, we made mice which lack the IPAS-specific 16th exon. In homozygous mice (IPAS^16Δ/16Δ^), endogenous IPAS loses its pro-apoptotic activity. The mutant mice were fully viable, fertile, apparently normal in appearance and behavior, and the expression of other HIF-3α splicing variants appeared to be normal. Acute administration of MPTP to IPAS^16Δ/16Δ^ mice caused a modest decrease in the number of TH-positive neurons, whereas MPTP significantly reduced TH-positive neurons in WT littermates. These results demonstrate that the pro-apoptotic activity of IPAS is involved in MPTP-dependent degeneration of dopaminergic neurons in the SNpc.

Some studies demonstrate that MK2-deficient mice are resistant to MPTP-induced neurodegeneration in the SNpc when compared with WT mice [72] and that the selective activation of p38 signaling by MPTP treatment occurs in dopaminergic neurons within the SNpc [73]. From these findings, we speculate that involvement of p38/MK2 signaling in neural death may be derived, at least partly, from the pro-apoptotic activity of IPAS by phosphorylation IPAS Ser^184^ in the SNpc of mice (Figure 4B). Next, expression levels of IPAS in the SNpc neurons of patients with sporadic PD were investigated [36]. Formalin-fixed, paraffin-embedded sections of the midbrain of six patients and six neurologically normal control individuals were analyzed by immunohistochemistry. The intracellular distribution of IPAS was similar to that observed in the TH-positive neurons of MPTP-treated mice. The intensity of IPAS immunostaining was significantly greater in the neurons of sporadic PD patients versus control individuals. These results suggest that IPAS contributes to the neurodegeneration of dopaminergic neurons in the SNpc of patients with sporadic PD as well as MPTP-treated mouse PD models (Figure 4B) [36].

## 4. Conclusions

In this review, we first focus on well-characterized PINK1 phosphorylation motifs. We also describe our findings for Parkin-Tg expression in mnd2 mouse models. Although Parkin protein levels in the striatum are decreased in mnd2 mice, the Parkin transgene did not rescue the neurodegenerative disorders in mnd2 mice. Recently, new mutations of CHCHD2 have been identified in familial and sporadic PD cases [11]. Although CHCHD2 primarily localizes to the intermembrane spaces of mitochondria, a pathogenic point mutation of CHCHD2 induces the translocation of CHCHD2 into cytosol where it aggregates with α-synuclein [74]. Under these conditions, the possibility exists that Parkin-Tg can prevent neurodegeneration by inhibiting aggregation of CHCHD2 and α-synuclein. Our Parkin-Tg mice are a powerful tool for elucidating these questions. Third, we describe our findings regarding IPAS, a newly identified target of the PINK1/Parkin pathway. IPAS is a bi-organellar factor with roles in nuclear transcription repression and mitochondrial pro-apoptotic function. This conversion is controlled by its subcellular localization via binding to anchoring proteins, and post-translational modification by other signaling pathways, such as p38 MAP kinase/MK2 signaling. The next challenge is a finding of the nuclear/mitochondrial anchoring proteins for IPAS by which IPAS bi-organellar function is regulated. Experiments using knockout mice lacking these proteins will dramatically enhance our understanding of novel pathogenic mechanism in PD.

## Figures and Tables

**Figure 1 ijms-21-01202-f001:**
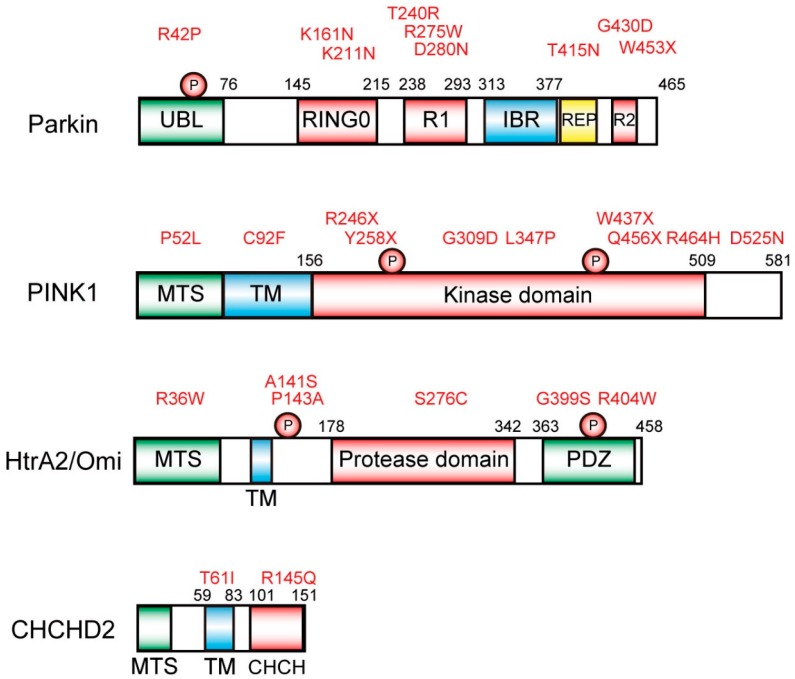
Structures of mitochondrial-localized proteins for genes associated with familial parkinsonism and their phosphorylation sites. The human Parkin, PINK1, HtrA2/Omi, and CHCHD2 proteins are drawn approximately to scale, with some pathogenic point mutations in red. Parkin contains a UBL domain at the N-terminus followed by three really-interesting-new-gene (RING) finger domains (R0-R2) separated by In-Between-RING fingers (IBR) and repressor element of Parkin (REP) domains. A phosphorylation site for PINK1 occurs at Ser^65^. PINK1 contains an MTS, transmembrane domain (TM), the kinase domain, and C-terminal sequences of uncertain function. Autophosphorylation sites are at Ser^228^ and Ser^402^. HtrA2/Omi consists of an MTS, TM, and a conserved catalytic trypsin-like serine protease domain, and a C-terminal PSD-95/Dlg/ZO-1 (PDZ) domain. Ser^142^ is phosphorylated by p38 in a PINK1-depedent manner and Ser^400^ is phosphorylated by CDK5. CHCHD2 has an MTS, TM, and CHCH domain. The CHCH domain consists of twin CX(9)C motifs and two disulfide bonds that stabilize the CHCH fold.

**Figure 2 ijms-21-01202-f002:**
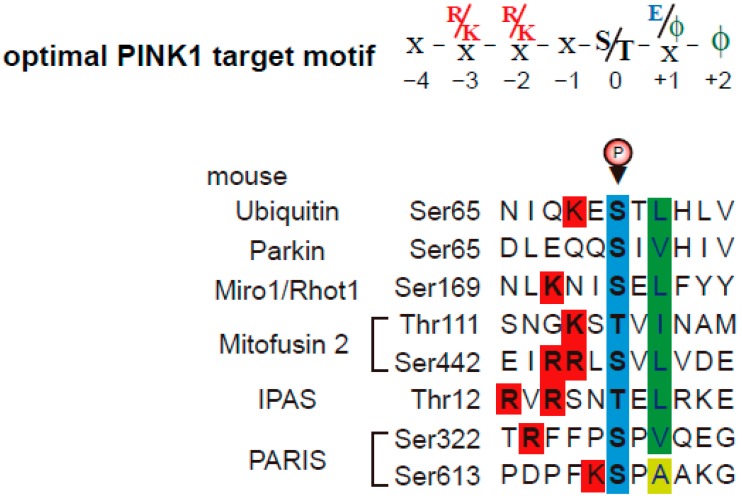
Alignment of amino acid sequences containing the phosphorylation sites of several PINK1 substrates. The known Ser or Thr residues phosphorylated by PINK1 (blue box) are indicated by an arrowhead with phosphorylation (P). Protein sequences of *Mus musculus* are presented. Hydrophobic amino acids (green box) and alanine (yellow box) found two residues (+2) after phosphorylation sites are shown. Hydrophobic amino acids or acidic amino acids at position +1 after the phosphorylation site are preferred. Basic amino acids found before the phosphorylation sites are shown (red box). Phosphorylation sites of drosophila MIC60/Mitofilin are not included because phosphorylation sites of mouse MIC60/Mitofilin corresponding to drosophila MIC60/Mitofilin cannot be identified using homology searches. From these observations, an optimal PINK1 target motif is given at the top using one-letter codes (ф: hydrophobic amino acids).

**Figure 3 ijms-21-01202-f003:**
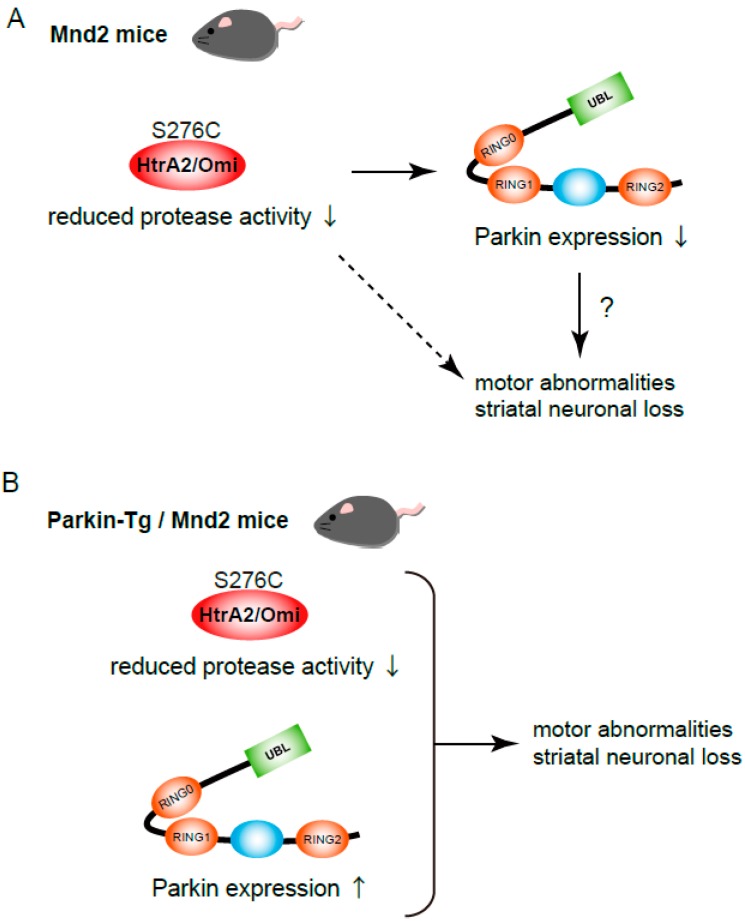
Neurodegeneration in mnd2 mutant mice is not prevented by Parkin-Tg. (**A**) Mnd2 mice have a non-functional missense mutation at Ser^276^ to Cys in the conserved catalytic trypsin-like serine protease domain of HtrA2/Omi which leads to motor neuron degeneration and selective loss of striatal neurons. In mnd2 mice, Parkin protein expression is low in the striatum. Parkin protein levels in the hippocampus are also decreased in mnd2 mice; the decrease is significantly weaker than that in striatum. It was unknown that low level of Parkin causes motor abnormalities and striatal neuronal loss in mnd2 mice before performing experiments with Parkin-Tg/ mnd2 mice (an arrow with a question mark). (**B**) Overexpression of Parkin by Parkin-Tg does not suppress PD phenotypes in mnd2 mice. Arrow and dashed arrow tracks indicate cause-and-effect relationships. Down-arrows and an up-arrow indicate low level and high level, respectively.

**Figure 4 ijms-21-01202-f004:**
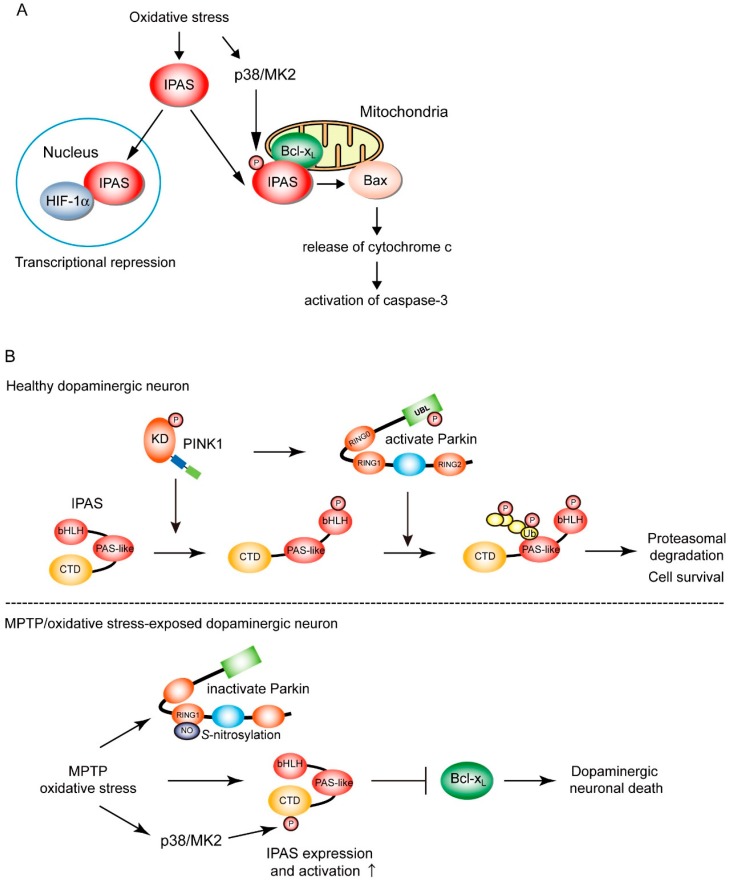
The PINK1-Parkin pathway regulates IPAS expression and inhibits neuronal death. (**A**) A model for the action of IPAS in transcriptional repression and apoptosis. Nuclear IPAS is bound to HIF-1α and inhibits hypoxic responses by inhibiting HIF-1α transcriptional activity. Mitochondrial IPAS is bound to Bcl-x_L_, Mcl-1, and A1a on the mitochondria and blocks the interaction of these proteins with Bax; thus activating of Bax, followed by release of cytochrome c and activation of caspase-3. p38-activated MK2 phosphorylates IPAS at Ser^184^, activating the pro-apoptotic functions of IPAS. IPAS has dual functions involved in the transcription repression of HIF-1α and cell death. (**B**) In healthy neurons, IPAS Thr^12^ is phosphorylated by PINK1 and then ubiquitinated by Parkin at lys^167–169^ in the PAS-like region, leading to the proteasomal degradation of IPAS. Phosphorylation of IPAS at Thr^12^ may induce a conformation change of IPAS to make the PAS-like region accessible to Parkin (top). The dopaminergic neurotoxin, MPTP, and other oxidative stresses strongly induce IPAS expression in the SNpc. Parkin is inhibited by *S*-nitrosylation or its ligase-dead pathologic mutation in PD. p38-activated MK2 may phosphorylate IPAS at Ser^184^ under these conditions. These events activate IPAS pro-apoptotic activity which leads to dopaminergic neuronal death in the SNpc (bottom). KD: kinase domain. Arrow tracks indicate signal transduction or IPAS translocation to nucleus/mitochondria. An up-arrow indicates high level of expression and activation.

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
