# Peer review of "Mitochondrial E3 Ubiquitin Ligase Parkin: Relationships with Other Causal Proteins in Familial Parkinson’s Disease and Its Substrate-Involved Mouse Experimental Models"

_ijms, 2020, doi:10.3390/ijms21041202_

Round 1

Reviewer 1 Report

After changes introduced as answers to the reviewers, the manuscript IJMS-714590V11 could be published in its present form.

Author Response

Point-by-point Responses

Responses to comment by Reviewer #1

Comment: After changes introduced as answers to the reviewers, the manuscript IJMS-714590V11 could be published in its present form.

Response:

We greatly appreciate this high evaluation. We have revised our manuscript in accordance with the reviewer’s comments below.

Responses to comment by Reviewer #2

Comment: As a general comment, despite this review article aims to discuss the most recent findings, it fails from reaching its goals since the most part of the cited references are older than 5-10 years ago. This is definitely not an up-to-date review article.

Response:

We appreciate this kind comment. Our review article aims to 1) update PINK1 phosphorylation motifs and 2) discuss our findings and their related studies involved in Parkinson’s disease. In accordance with the comments, we deleted the word “latest” and in abstract (Page 1 line 22) and modified abstract and conclusion (Page 1 line 21-23, Page 10 line 399). And we added recent works involved in our studies (Page 7 line 238, 270-271, 279).

Comment: There are a number of references that are erroneously reported to be recent. In my opinion, it is not possible to consider as recent those findings published in 2008, 2010, 2011, 2012, or 2013. As an instance, some examples are (but not limited to): Page 4, line 132; page 7, line 223; page 7, line 251; page 10, line 338. Importantly, a significant part of the findings reported in this manuscript is based on research articles produced by the authors in 2015, 2010, 2009, and 2011. This fact severely undermine the putative novelty of the review article.

Response:

We appreciate this kind comment. In accordance with this comments, we deleted the word “recent/recently” in revised manuscript (Page 4 line 144, Page 7 line 238, 265, Page 10 line 385, 408).

Our new papers involved in Parkinson’s disease are in preparation. In this invited review, we summarize our previous findings before new papers come.

Comment: The work proposed by the authors results to be extremely self-referential, since it mainly focuses the attention of experimental works produced by the authors themselves during the past years. The review article is clear and well written, however its contribution to the specific field of research is very weak. A review article should summarize and critically discuss a plethora of experimental data in the context of a specific topic. I really miss the point of proposing a review article when the major findings discussed in this manuscript are restricted to few articles published by the same authors. Literature data is very limited, and the natural consequence of this issue consists in a weak and extremely self-referential article.

Response:

We appreciate this constructive comment. In accordance with this suggestion, we added some other works involved in our studies (Page 7 line 238, 270-271, 279). Actually, the studies about IPAS are now limited and we can’t completely discuss them by comparing our data with other data, particularly in the role of apoptosis and Parkinson’s disease. We would like researchers to know this forgotten interesting protein in this planned review.

Responses to comment by Reviewer #3

Comment: The article entitled "Mitochondrial E3 ubiquitin ligase Parkin: relationships with other causal proteins in familial PD and its substrate-involved mouse experimental models” by Torii et al. reviews several genes and associated proteins, for which mutations at the gene level, or modifications of proteins (in particular post-traditional phosphorylation or ubiquitination) favor mechanisms of neuronal degeneration, via the induction of apoptosis by inhibition of mitochondrial autophagy. These mechanisms are at the origin of familial forms of Parkinson's disease.

The review is mainly focused on the results of this team. But it has the merit of making an updated point on genes, and their proteins, of major importance for the development of PD.

Response:

We greatly appreciate this high evaluation.

Comment:

Minor

L23: substrates, involved

L42: Exposure to other neurotoxic substances, including 6-OHDA, or the pesticides rotenone and paraquat…

L43: They produce effects similar to…

L45: acute and subacute.

L116: inhibitory PAS (Per/Arnt/Sim) domain protein (PAS). Remove from L.235

Response:

We appreciate this kind comment. In accordance with this suggestion, we modified our manuscript (Page 1 line 24-25, Page 2 line 47, 48, 50, Page 3 line 126).

Reviewer 2 Report

As a general comment, despite this review article aims to discuss the most recent findings, it fails from reaching its goals since the most part of the cited references are older than 5-10 years ago. This is definitely not an up-to-date review article.

There are a number of references that are erroneously reported to be recent. In my opinion, it is not possible to consider as recent those findings published in 2008, 2010, 2011, 2012, or 2013. As an instance, some examples are (but not limited to): Page 4, line 132; page 7, line 223; page 7, line 251; page 10, line 338. Importantly, a significant part of the findings reported in this manuscript is based on research articles produced by the authors in 2015, 2010, 2009, and 2011. This fact severely undermine the putative novelty of the review article.

The work proposed by the authors results to be extremely self-referential, since it mainly focuses the attention of experimental works produced by the authors themselves during the past years. The review article is clear and well written, however its contribution to the specific field of research is very weak. A review article should summarize and critically discuss a plethora of experimental data in the context of a specific topic. I really miss the point of proposing a review article when the major findings discussed in this manuscript are restricted to few articles published by the same authors. Literature data is very limited, and the natural consequence of this issue consists in a weak and extremely self-referential article. 

Author Response

Responses to comment by Reviewer #2

Comment: As a general comment, despite this review article aims to discuss the most recent findings, it fails from reaching its goals since the most part of the cited references are older than 5-10 years ago. This is definitely not an up-to-date review article.

Response:

We appreciate this kind comment. Our review article aims to 1) update PINK1 phosphorylation motifs and 2) discuss our findings and their related studies involved in Parkinson’s disease. In accordance with the comments, we deleted the word “latest” and in abstract (Page 1 line 22) and modified abstract and conclusion (Page 1 line 21-23, Page 10 line 399). And we added recent works involved in our studies (Page 7 line 238, 270-271, 279).

Comment: There are a number of references that are erroneously reported to be recent. In my opinion, it is not possible to consider as recent those findings published in 2008, 2010, 2011, 2012, or 2013. As an instance, some examples are (but not limited to): Page 4, line 132; page 7, line 223; page 7, line 251; page 10, line 338. Importantly, a significant part of the findings reported in this manuscript is based on research articles produced by the authors in 2015, 2010, 2009, and 2011. This fact severely undermine the putative novelty of the review article.

Response:

We appreciate this kind comment. In accordance with this comments, we deleted the word “recent/recently” in revised manuscript (Page 4 line 144, Page 7 line 238, 265, Page 10 line 385, 408).

Our new papers involved in Parkinson’s disease are in preparation. In this invited review, we summarize our previous findings before new papers come.

Comment: The work proposed by the authors results to be extremely self-referential, since it mainly focuses the attention of experimental works produced by the authors themselves during the past years. The review article is clear and well written, however its contribution to the specific field of research is very weak. A review article should summarize and critically discuss a plethora of experimental data in the context of a specific topic. I really miss the point of proposing a review article when the major findings discussed in this manuscript are restricted to few articles published by the same authors. Literature data is very limited, and the natural consequence of this issue consists in a weak and extremely self-referential article.

Response:

We appreciate this constructive comment. In accordance with this suggestion, we added some other works involved in our studies (Page 7 line 238, 270-271, 279). Actually, the studies about IPAS are now limited and we can’t completely discuss them by comparing our data with other data, particularly in the role of apoptosis and Parkinson’s disease. We would like researchers to know this forgotten interesting protein in this planned review.

Reviewer 3 Report

ijms-714590

The article entitled "Mitochondrial E3 ubiquitin ligase Parkin: relationships with other causal proteins in familial PD and its substrate-involved mouse experimental models” by Torii et al. reviews several genes and associated proteins, for which mutations at the gene level, or modifications of proteins (in particular post-traditional phosphorylation or ubiquitination) favor mechanisms of neuronal degeneration, via the induction of apoptosis by inhibition of mitochondrial autophagy. These mechanisms are at the origin of familial forms of Parkinson's disease.

The review is mainly focused on the results of this team. But it has the merit of making an updated point on genes, and their proteins, of major importance for the development of PD.

Minor

L23: substrates, involved

L42: Exposure to other neurotoxic substances, including 6-OHDA, or the pesticides rotenone and paraquat…

L43: They produce effects similar to…

L45: acute and subacute.  

L116: inhibitory PAS (Per/Arnt/Sim) domain protein (PAS). Remove from L.235

Author Response

Responses to comment by Reviewer #3

Comment: The article entitled "Mitochondrial E3 ubiquitin ligase Parkin: relationships with other causal proteins in familial PD and its substrate-involved mouse experimental models” by Torii et al. reviews several genes and associated proteins, for which mutations at the gene level, or modifications of proteins (in particular post-traditional phosphorylation or ubiquitination) favor mechanisms of neuronal degeneration, via the induction of apoptosis by inhibition of mitochondrial autophagy. These mechanisms are at the origin of familial forms of Parkinson's disease.

The review is mainly focused on the results of this team. But it has the merit of making an updated point on genes, and their proteins, of major importance for the development of PD.

Response:

We greatly appreciate this high evaluation.

Comment:

Minor

L23: substrates, involved

L42: Exposure to other neurotoxic substances, including 6-OHDA, or the pesticides rotenone and paraquat…

L43: They produce effects similar to…

L45: acute and subacute.

L116: inhibitory PAS (Per/Arnt/Sim) domain protein (PAS). Remove from L.235

Response:

We appreciate this kind comment. In accordance with this suggestion, we modified our manuscript (Page 1 line 24-25, Page 2 line 47, 48, 50, Page 3 line 126).

Round 2

Reviewer 2 Report

Despite this review article still has some limitations, the authors did their best to address the reviewer's comment. The revised version of this manuscript results to be slightly improved.